# Human mitochondrial ferritin exhibits highly unusual iron-O$_2$ chemistry distinct from that of cytosolic ferritins

Justin M. Bradley [1], Zinnia Bugg[1], Jacob Pullin[2,3], Geoffrey R. Moore[1], Dimitri A. Svistunenko[2] & Nick E. Le Brun [1]✉

Ferritins are ubiquitous proteins that function in iron storage/detoxification by catalyzing the oxidation of Fe$^{2+}$ ions and solubilizing the resulting Fe$^{3+}$-oxo mineral. Mammalian tissues that are metabolically highly active contain, in addition to the widespread cytosolic ferritin, a ferritin that is localized to mitochondria. Mitochondrial ferritin (FtMt) protects against oxidative stress and is found at higher levels in diseases associated with abnormal iron accumulation, including Alzheimer's and Parkinson's. Here we demonstrate that, despite 80% sequence identity with cytosolic human H-chain ferritin, Fe$^{2+}$ oxidation at the catalytic diiron ferroxidase center of FtMt proceeds via a distinct mechanism. This involves a mixed-valent ferroxidase center (MVFC) that is readily detected under the O$_2$-limiting conditions typical of mitochondria, and formation of a radical on a strictly conserved Tyr residue (Tyr34) that is key for the activation of O$_2$ and stability of the MVFC. The possible origin of the mechanistic differences exhibited by the highly-related human mitochondrial and cytosolic H-chain ferritins is explored.

Iron is an essential micronutrient and is contained in many enzymes that catalyze important biological functions[1–3]. However, in the presence of O$_2$, iron is readily oxidized to the Fe$^{3+}$ oxidation state, forming insoluble ferric (oxy)hydroxide minerals at pH > 3[4]. Furthermore, interconversion between the Fe$^{2+}$ and Fe$^{3+}$ oxidation states catalyzes the formation of the highly toxic hydroxyl radical from superoxide and peroxide anions generated as by-products of aerobic respiration[5]. Therefore, oxygenation of the atmosphere during the Paleoproterozoic era presented the dual challenges of poor bioavailability and potential toxicity for the utilization of iron in biological systems. In most organisms, members of the ferritin superfamily are important in overcoming these challenges[6].

Ferritins were first identified over 85 years ago[7] and all examples identified in eukaryotes consist of 24 protein subunits assembled into a rhombic dodecahedral cage surrounding an 80 Å diameter interior cavity. Central to ferritin activity is a diiron catalytic site termed the ferroxidase center (FC)[8–10], which is key for the detoxification/storage

of iron as an insoluble ferric mineral in the protein's internal cavity[6,11,12]. This activity limits the intracellular concentration of the potentially toxic Fe$^{2+}$ to low micromolar levels[13–15], required to ensure correct metallation of iron-containing proteins[16]. Iron stored within the mineral core may be mobilized under conditions of iron limitation in order to maintain the concentration of the labile iron pool[17].

Ferritins isolated from the cytosol or the serum of animals are heteropolymers of isostructural H-chains, which contain an FC, and L-chains that do not[18]. Proportions of H- and L-chains vary depending on the tissue of origin. For example, ferritin isolated from the liver is rich in L-chain, while that isolated from the heart is rich in H-chain. This variation in composition is believed to be linked to function, where H-chains promote rapid Fe$^{2+}$ oxidation, resulting in detoxification, and L-chains promote mineral core nucleation, facilitating iron storage[8,19].

H- and L-chains were thought to be the only ferritins expressed by mammals until the report in 2001 of an intronless gene, located on human chromosome 5q23.1, which encodes a ferritin with an

[1]Centre for Molecular and Structural Biochemistry, School of Chemistry, Pharmacy and Pharmacology, University of East Anglia, Norwich, UK. [2]School of Life Sciences, University of Essex, Colchester, UK. [3]Present address: The John Innes Centre, Norwich, UK. ✉e-mail: n.le-brun@uea.ac.uk

N-terminal mitochondrial targeting sequence[20]. Expression of this homopolymer mitochondrial ferritin (FtMt) is not uniform throughout all tissues and, unlike the cytosolic proteins, is not under the control of an Iron Responsive Element in the ferritin mRNA[18,21,22]. Consistent with this, the expression levels of mouse FtMt were found to correlate with mitochondrial density rather than iron levels[23]. FtMt is only detected in the cells of tissues with high metabolic activity, such as the testis, heart, brain, and kidney, and the protein is not expressed in the liver or splenocytes, where cytosolic ferritins are highly expressed for iron storage[24]. Despite this, the protein has attracted increasing attention, as its misregulation is associated with many disease states, particularly neurological disorders[25–27]. While a high-resolution X-ray structure[22] and some initial studies of iron mineralization[28] were published shortly after the protein's discovery, the focus of most studies has been the physiological consequences of misregulation of FtMt expression. Thus, the biochemical and biophysical properties of the protein and the mechanism of $Fe^{2+}$ oxidation/mineralization remain relatively poorly characterized.

Following cleavage of the mitochondrial targeting sequence, FtMt has 80% sequence identity to the H-chain of the cytosolic protein (HuHF), including conservation of all residues that act as ligands to the FC. Despite this, marked differences have been reported in the iron-binding and oxidation properties of the two proteins[22,28]. The mechanism of HuHF is well established: following the binding of two $Fe^{2+}$ ions to the FC of the iron-free (apo-) protein, reaction with $O_2$ generates a di-$Fe^{3+}$ peroxo (DFP) species[29] that is hydrolyzed to yield a metastable di-$Fe^{3+}$ oxo form of the FC. Displacement of $Fe^{3+}$ by further equivalents of $Fe^{2+}$ both initiates mineral core formation and regenerates the di-$Fe^{2+}$ form of the FC, allowing multiple turnovers of the site and nucleation/growth of the iron mineral (Fig. 1).

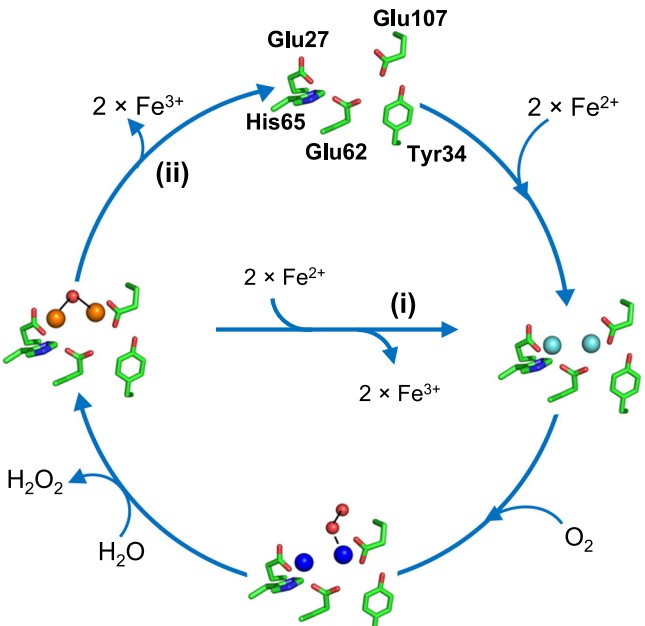

**Fig. 1 | Mechanism of $Fe^{2+}$ oxidation at cytosolic ferritin ferroxidase centers (FCs).** Starting at the top, two equivalents of $Fe^{2+}$ bind at the vacant (apo) FC to give di-$Fe^{2+}$ FC (right). Transfer of a single electron from each onto $O_2$ results in the formation of a di-$Fe^{3+}$ peroxo intermediate (bottom) that is hydrolyzed to give a metastable di-$Fe^{3+}$ oxo species and peroxide (left). In pathway (i) $Fe^{3+}$ is then displaced by incoming $Fe^{2+}$ substrate, regenerating the di-$Fe^{2+}$ form of the FC and leading to the accumulation of a ferrihydrite-like mineral in the interior of the protein. In the absence of further $Fe^{2+}$ substrate (pathway (ii)), the apo form of the FC is regenerated by the slower loss of $Fe^{3+}$, into the interior of the protein, where it also contributes to the accumulation of a mineral core. $Fe^{2+}$ ions are indicated in cyan, $Fe^{3+}$ ions are in brown, and the DFP is in royal blue.

Displacement of the oxidized product of the HuHF FC reaction is slow in comparison to binding and oxidation of $Fe^{2+}$, meaning that initial turnover of the FC is kinetically distinct from the formation of the mineral core[11]. The former process in FtMt, characterized by aerobic titration of $Fe^{2+}$ up to the point of saturating binding at FC sites, revealed unexpected features, including discontinuities at a ratio of 24 $Fe^{2+}$/FtMt (i.e. 50% of that required to saturate FC binding) in parameters indicative of binding and/or oxidation of $Fe^{2+}$ to FCs. Equivalent data for HuHF showed no evidence of such discontinuities, and the reported rate of the subsequent iron core mineralization process in FtMt was approximately 50% of that for HuHF. These observations led to the proposal that only half of the catalytic FCs of FtMt are active.

The majority of the amino acid residues that differ between HuHF and the mature FtMt peptide sequences are located at the N and C termini, or the outer surface of the protein cage[22]. A notable exception is residue Ser144 (FtMt numbering, post-maturation), the equivalent residue in HuHF being alanine. In fact, the sequences of animal H-chain ferritins display considerable variation at this position, whilst this residue is invariably Ser in their mitochondrial equivalents (Fig. S1). FtMt variant S144A was reported to have greater FC activity and faster iron mineralization kinetics than the wild-type protein, leading to the suggestion that negative cooperativity of $Fe^{2+}$-binding to FtMt FCs, in part mediated by Ser144, is responsible for its unusual iron-binding and oxidation properties[28].

Recently, SynFtn, a ferritin from a coastal-dwelling cyanobacterium that also contains Ser at the position equivalent to Ser144 in FtMt, was characterized[30]. This residue was identified as an important factor in directing the mechanism of iron oxidation in SynFtn via an unprecedented pathway, involving a tyrosyl radical and an unusual mixed-valent $Fe^{2+}$/$Fe^{3+}$ intermediate form of the FC (MVFC)[31]. The site of tyrosyl radical formation was identified as Tyr40 (SynFtn numbering), a residue that is conserved in FtMt (Tyr34). Formation of the MVFC involved oxidation of only 50% of the $Fe^{2+}$ at FC sites, potentially linking the behavior of SynFtn and the early reports of FtMt activity. Therefore, we sought to characterize the intermediates formed during the oxidation of $Fe^{2+}$ catalyzed by FtMt.

## Results

### Wild-type FtMt-catalyzed oxidation of $Fe^{2+}$ under air-saturated conditions results in a protein-based radical, but no MVFC

Addition of $Fe^{2+}$ ions to apo FtMt in air-saturated buffer resulted in a rapid initial reaction, corresponding to the oxidation of two $Fe^{2+}$ ions at each FC, followed by a slower reaction due to the subsequent formation of a mineral core. The oxo-coordinated $Fe^{3+}$-containing species generated at the FC and the ferrihydrite-like mineral core both give rise to broad absorbance features centered at 340 nm with extinction coefficients of approximately 2000 $M^{-1} cm^{-1}$ at this wavelength[32]. Thus, the rate of increase in 340 nm absorbance is proportional to the rate of $Fe^{2+}$ oxidation and hence protein activity. Figure 2a shows the increase in 340 nm absorbance following the addition of 400 equivalents of $Fe^{2+}$ to wild-type FtMt. Two clear phases were observed, the slower phase representing mineral core formation following an initial jump due to turnover of the FC.

Stopped-flow absorbance measurements at 340 nm following the addition of increasing amounts of $Fe^{2+}$ (up to 96 per 24mer) to apo FtMt (Fig. 2b) revealed the FC reaction in full. A single phase was observed at lower $Fe^{2+}$ loading, saturating at ~48 $Fe^{2+}$ per cage, and with a $\Delta A_{340 nm}$ amplitude consistent with oxidation of all $Fe^{2+}$ to $Fe^{3+}$ (Fig. 2c). This was unexpected given the earlier report of activity in only half of the FtMt FCs[28]. At higher loadings, a biphasic response was observed, with the amplitude of the slower phase increasing in proportion to the amount of $Fe^{2+}$ added beyond that needed to saturate the initial rapid phase. Fitting of the data to one or two exponential functions (as needed) yielded effective first-order rate constants for

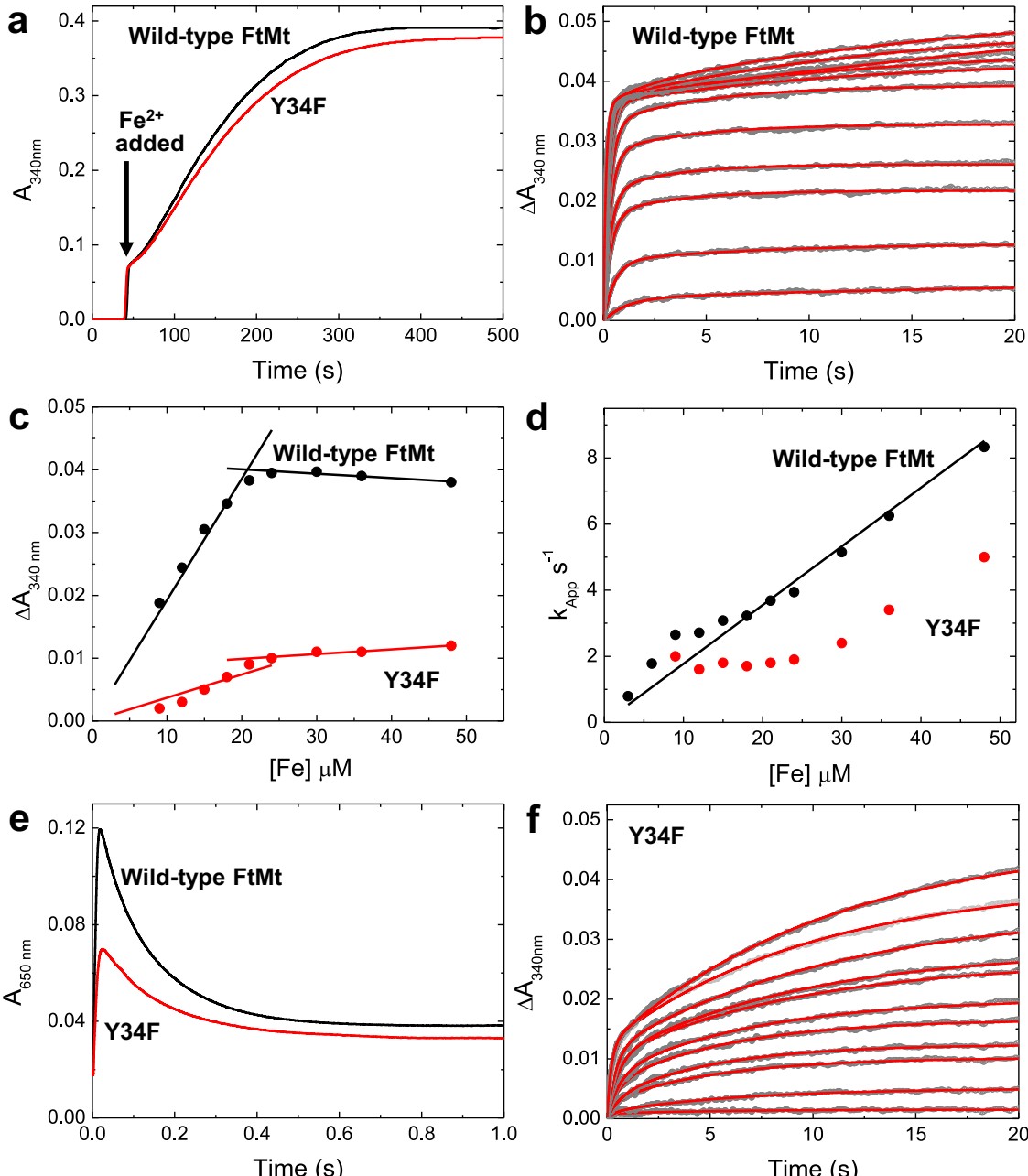

**Fig. 2 | Iron oxidation and mineralization by FtMt. a** The increase in 340 nm absorbance as a function of time following the aerobic mixing of wild-type (black trace) and variant Y34F (red trace) FtMt with 400 equivalents of $Fe^{2+}$. **b** The increase in 340 nm absorbance as a function of time following the aerobic mixing of wild-type FtMt with equal volumes of solutions containing 6–96 equivalents of $Fe^{2+}$. Data are shown as gray circles; red traces represent fits to either mono- or bi-exponential functions. **c** The amplitude of the increase in 340 nm absorbance during the rapid phase of $Fe^{2+}$ oxidation by wild-type (black) and variant Y34F (red) FtMt as a function of the concentration of added $Fe^{2+}$. **d** The dependence of the apparent first-order rate constant for $Fe^{2+}$ oxidation as a function of initial $Fe^{2+}$ concentration for wild-type (black) and variant Y34F (red) FtMt. The black line represents the best fit through the origin to the data for the wild-type protein. **e** The change in 650 nm absorbance as a function of time following the aerobic mixing of wild-type (black trace) and variant Y34F (red trace) FtMt with 48 equivalents of $Fe^{2+}$, reporting the formation and subsequent decay of the di-$Fe^{3+}$ peroxo intermediate. **f** as (**b**) but for variant Y34F FtMt. Source data are provided as a Source Data file.

these processes, that for the rapid phase are plotted as a function of $Fe^{2+}$ concentration in Fig. 2d. The dependence was approximately linear, indicating that $Fe^{2+}$ binding to the FC is rate-limiting under the conditions employed and allowing the calculation of a second order rate constant of $1.8 \times 10^5 \, M^{-1} \, s^{-1}$ for this process, approximately 50% of the value of $3.8 \times 10^5 \, M^{-1} \, s^{-1}$ determined for HuHF (Fig. S2)[28]. Parameters describing the rapid oxidation of $Fe^{2+}$ by FtMt FCs extracted from these data are listed in Table S1.

In contrast to early studies of FtMt conducted at pH 7.0[28], the above data indicated the rapid oxidation of 48 equivalents of $Fe^{2+}$ per protein

cage. Since this represents the total number of iron-binding sites located within the FCs of the protein, we expected to find that wild-type FtMt directly forms the commonly observed diferric-peroxo (DFP) intermediate species during $O_2$-driven activity. DFP species have a characteristic absorbance between 600 and 650 nm with an extinction coefficient that varies with the coordination geometry of the peroxide relative to the iron[11,29,33–36]. Stopped-flow measurements at 650 nm confirmed that a DFP intermediate formed in FtMt (Fig. 2e), and that it subsequently decayed, most likely to a metastable μ-oxo bridged diferric species, as observed for all eukaryotic H-chain ferritins studied to date[18].

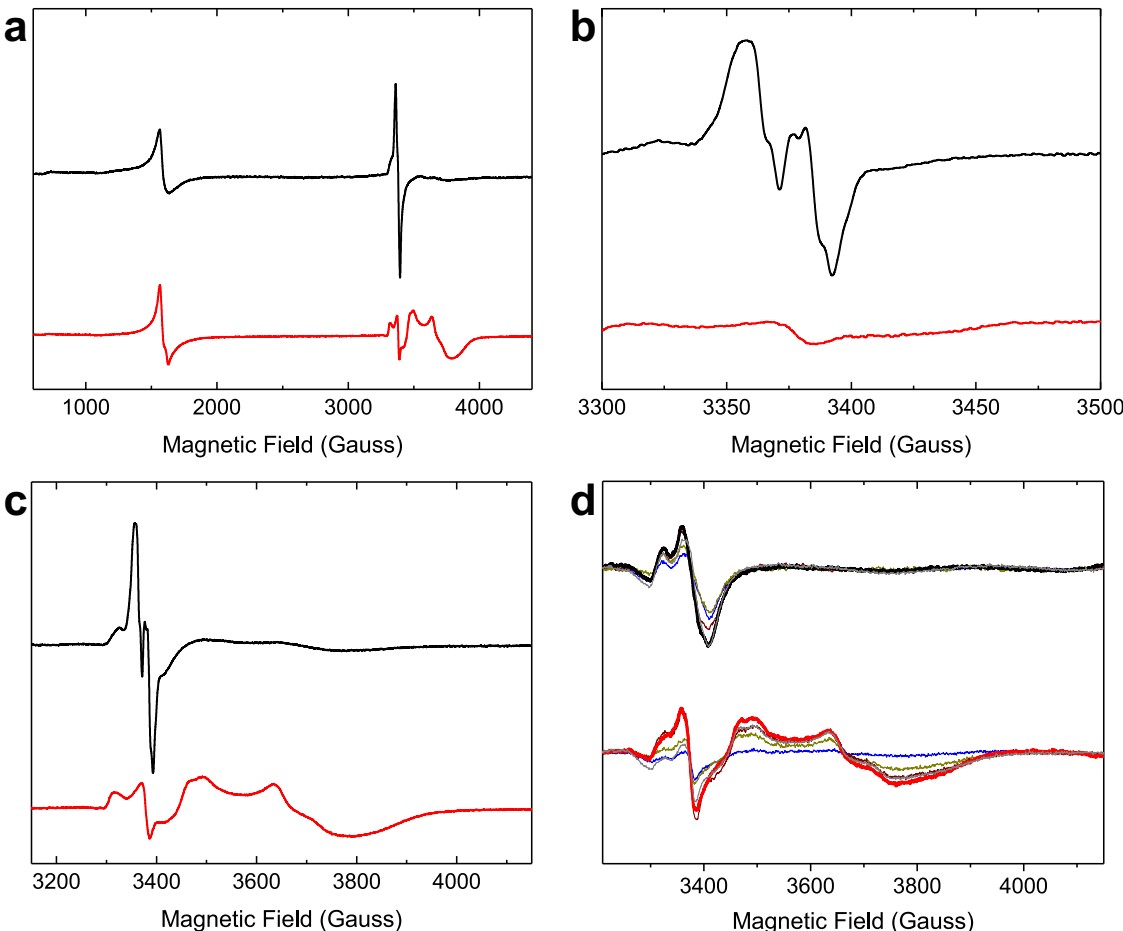

**Fig. 3 | EPR spectra of the paramagnetic species formed during Fe²⁺ oxidation by FtMt. a** Spectra recorded at microwave power of 3.19 mW over the field range 600–4200 Gauss for 4.2 μM wild-type (black trace) and variant Y34F (red trace) FtMt frozen approximately 10 s following aerobic mixing with 72 equivalents of Fe²⁺. **b** as in (**a**) but spectra recorded over the field range 3300–3500 Gauss at microwave power 0.05 mW. **c** as (**a**) but spectra recorded over the field range 3200–4150 Gauss. **d** as (**c**) but with freezing times for the wild-type protein of 45 ms (blue), 125 ms (yellow), 245 ms (brown), 445 ms (heavy black trace with maximum radical intensity), or 10 s (gray) and freezing times for variant Y34F of 45 ms (blue), 245 ms (yellow), 445 ms (brown), 3 s (heavy red trace with maximum mixed-valent ferroxidase center intensity) or 10 s (gray). Source data are provided as a Source Data file.

The absorbance data above imply that there is no accumulation of a MVFC intermediate in FtMt. Electron paramagnetic resonance (EPR) spectroscopy was used to further investigate this. Figure 3a shows the EPR spectrum of wild-type FtMt frozen approximately 10 s after mixing with 72 equivalents of Fe²⁺ in air-saturated buffer. The spectrum contained a signal at $g = 4.3$, which is commonly observed during ferritin-catalyzed Fe²⁺ oxidation, indicating the presence of high-spin Fe³⁺ in a low symmetry environment. A second signal was observed at approximately $g = 2$, typical of a protein-based radical located on an oxidized tyrosine residue. This is shown over a narrower field range in Fig. 3b. Very low intensity signals below $g = 2$ suggested the possible presence of a minor amount of MVFC (Fig. 3c), though this was variable from sample to sample and never amounted to more than approximately 1% of protein monomer concentration.

The above data provide no indication of the accumulation of significant amounts of a MVFC. Nevertheless, rapid freeze quench (RFQ) was used to assess freezing times between 50 ms and 10 s to rule out the possibility that observable quantities of a MVFC species form and decay in wild-type FtMt during the first several seconds of the reaction. The intensity of the protein-based radical signal increased with freezing time up to 445 ms and then remained constant over the following 10 s. Again, the spectra showed no evidence of significant MVFC formation (Fig. 3d). Thus, the data indicate that a Tyr protein

radical species is formed during Fe²⁺ oxidation at the FC, but significant amounts of a MVFC are not.

## Tyr34 is the site of radical formation on FtMt and plays an important role in initial Fe²⁺ oxidation, but not mineralization

All H-chain type ferritins contain a strictly conserved Tyr residue lying close to the FC, which has been shown in several cases to play an important role in Fe²⁺ oxidation/mineralization[17,36–38]. In FtMt, this is Tyr34, and so the effect of a Y34F substitution on FtMt activity was investigated by measuring the rate of increase in absorbance at 340 or 650 nm following aerobic addition of Fe²⁺ compared with equivalent data for the wild-type protein. Figure 2a shows the increase in 340 nm absorbance due to the formation of the mineral core following the addition of 400 equivalents of Fe²⁺ to Y34F FtMt was almost identical to that of the wild-type protein, indicating that substitution of Tyr34 had no significant effect on the rate at which FtMt can mineralize iron. We note that the equivalent substitution in HuHF resulted in an approximately 50% decrease in mineralization activity (Fig. S2a).

In contrast to mineral core formation, the initial Fe²⁺ oxidation reaction was very different for the Y34F variant compared to wild-type FtMt, see Fig. 2f. Not only was the rate of the rapid oxidation phase reduced in the variant, but the amplitude of this phase for Y34F FtMt saturated at ~30% of that for wild-type FtMt (Fig. 2c). Furthermore, the rate constant associated with the rapid phase observed for Y34F was

independent of $Fe^{2+}$ concentration below 24 μM (48 Fe per cage), at which point it began to increase linearly (Fig. 2d), demonstrating that, at low $Fe^{2+}$ concentration, iron binding to Y34F is not the rate-limiting step of the FC-catalyzed oxidation reaction. Equivalent data for the Y34F variant of HuHF demonstrated that rapid oxidation is abolished by this substitution (Fig. S2).

The rate of formation of the DFP intermediate by the FtMt Y34F variant, indicative of the rate of $Fe^{2+}$ oxidation at the FC, was only 65% of that observed for the wild-type FtMt, with apparent first-order rate constants of 70 $s^{-1}$ and 110 $s^{-1}$, respectively, when mixing protein and $Fe^{2+}$ under aerobic conditions (Fig. 2e). The rate of FC-catalyzed $Fe^{2+}$ oxidation was also monitored using a different experimental arrangement in which equal volumes of aerobic buffer and FtMt anaerobically pre-incubated with 48 $Fe^{2+}$ per cage were mixed (Fig. S3). This mixing protocol eliminated any effect due to different rates of $Fe^{2+}$ binding to the FC of wild-type and variant Y34F, demonstrating that the intrinsic rate of $Fe^{2+}$ oxidation is 35-fold slower in the variant protein. Consistent with this, the rate of DFP formation is severely impaired in the variant protein, such that detectable quantities of this intermediate only accumulate at ≥30 °C. Even under conditions of aerobic mixing, the maximum increase in $A_{650\,nm}$ observed for variant Y34F was only approximately 50% of that of the wild-type FtMt. This suggests either a change in the absorbance properties of the DFP due to a different binding mode of peroxide, or that the substitution impacted the extent to which this intermediate was generated (Fig. 2e). Given the reduction in 340 nm absorbance amplitude associated with the rapid phase of the FC reaction (Fig. 2b), this was most likely a consequence of impaired DFP formation. EPR spectroscopy was employed to determine whether this was due to the formation of a MVFC species in competition with the DFP intermediate.

## Formation of a MVFC in Y34F FtMt during aerobic $Fe^{2+}$ oxidation

The spectrum of Y34F FtMt frozen approximately 10 s after mixing with 72 equivalents of $Fe^{2+}$ in air-saturated buffer contained the $g = 4.3$ $Fe^{3+}$ feature observed for the wild-type protein, but at lower intensity (Fig. 3a). The signal in the $g = 2$ region was unlike that observed for the wild-type protein, in that the tyrosyl radical signal was replaced by a signal typical of oxidative damage due to peroxidation of amino acid sidechains (ROO·), see Fig. 3b for a narrower field range.

In addition to the signals described above, the spectrum of variant Y34F also contained a rhombic signal with all g-values below 2 (Fig. 3a, c), characteristic of previously reported MVFCs[12,30,39,40]. Integration of the MVFC envelope indicated a concentration of this species of approximately 30 μM (equating to ~30% of subunit concentration). The kinetics of MVFC formation were investigated using RFQ EPR, where freezing times between 50 ms and 10 s were accessed (Fig. 3d). The data revealed the formation of a MVFC, together with ROO· species, over the first 3 s of the reaction, with the majority formed in the first 500 ms. This is similar to the kinetics of DFP formation detected by stopped-flow absorbance, suggesting that in this variant the two species form by competing direct oxidation pathways. The ROO· species decayed over the following 7 s, whilst the MVFC signal remained stable with a maximum observed concentration of approximately 40 μM (Fig. 3d).

Thus, the combination of absorbance and EPR kinetic data demonstrated that the Y34F variant of FtMt forms a MVFC species directly during its reaction with excess $O_2$, but an equivalent species is not readily detected in the wild-type protein. Importantly, the EPR spectrum of a sample of Y34F HuHF frozen approximately 10 s after mixing with 72 equivalents of $Fe^{2+}$ in air-saturated buffer contained no evidence of MVFC formation (Fig. S4), indicating that the observation of a MVFC following substitution of Tyr34 by Phe is unique to FtMt and not common to H-chain ferritins.

## Formation of a MVFC in wild-type FtMt under limiting $O_2$

Assays of ferritin activity are usually conducted under conditions of excess oxidant, typically $O_2$, such that the reaction terminates when all added $Fe^{2+}$ has been oxidized to $Fe^{3+}$. To further probe the FtMt-catalyzed reaction between $Fe^{2+}$ and $O_2$, samples were prepared with insufficient $O_2$ to oxidize all the added $Fe^{2+}$. According to the reaction scheme of Fig. 1, the predicted result would be ferritin cages containing a mixture of di-$Fe^{3+}$-oxo and unreacted di-$Fe^{2+}$ FCs. Indeed, the EPR spectrum of a control sample of HuHF treated in this way was entirely consistent with this outcome (Fig. S4), with only trace amounts of MVFC present, most likely formed via displacement of $Fe^{3+}$ from di-$Fe^{3+}$ centers by trace unreacted $Fe^{2+}$ remaining in solution[12]. However, the EPR spectrum of wild-type FtMt clearly contained the features associated with formation of a MVFC, with approximately 5% of protein monomers containing an FC in this oxidation state, too great to plausibly be formed by trace unreacted $Fe^{2+}$ remaining in solution via the same mechanism as for HuHF (Fig. 4a). Several possibilities exist for how this oxidation state of the FtMt FC forms, including: direct oxidation to a MVFC intermediate that is trapped when there is insufficient $O_2$ to drive further oxidation to the di-$Fe^{3+}$ form, protein-mediated electron transfer between unreacted FCs and di-$Fe^{3+}$ FCs to yield 2 equivalents of the MVFC form[30], or dissociation of unreacted $Fe^{2+}$ from di-$Fe^{2+}$ FCs and subsequent displacement of $Fe^{3+}$ from a di-$Fe^{3+}$ FC.

Further EPR experiments were performed in which partial occupancy of FCs or increased concentration of unbound $Fe^{2+}$ in solution were deliberately introduced in the wild-type protein. The goal was to probe whether a lower $Fe^{2+}$ binding affinity of Y34F FtMt caused the observation of a MVFC in this protein. The data showed that this was not the case, see Fig. S4 for further details. Thus, we conclude that under excess $O_2$, any MVFC intermediate, generated by either electron transfer from partially occupied to fully oxidized FCs, or displacement of $Fe^{3+}$ by $Fe^{2+}$, does not accumulate. This suggested that the MVFC form of wild-type FtMt catalytic centers is only stable in the absence of $O_2$ and, unlike the equivalent MVFC form of SynFtn, reacts rapidly with $O_2$ to generate di-$Fe^{3+}$ sites. Absorbance-monitored stopped-flow mixing data support this conclusion. Wild-type FtMt was prepared in the MVFC state using higher protein and $Fe^{2+}$ concentrations of 20.8 μM and 1 mM, respectively, such that the typical ambient dissolved $O_2$ concentration (250 μM) was sufficient to oxidize only 50% of FC sites. A sample prepared in this way within a gas-tight syringe showed no increase in absorbance on mixing with anaerobic buffer, whilst mixing with buffer containing 250 μM dissolved $O_2$ resulted in an increase of 0.3 absorbance units within the deadtime of the instrument (Fig. S6).

Increased protein and $Fe^{2+}$ concentrations were also exploited to probe the relative rates of DFP and MVFC formation in wild-type FtMt under $O_2$-limited conditions. Slow-freeze experiments (Fig. S7) performed with equal volume mixing of protein and iron solutions (for equivalence with later RFQ experiments) demonstrated that significant MVFC formation occurred under these conditions. However, the time dependence of MVFC formation (Fig. 4b, c) measured by RFQ experiments showed that it did not begin to accumulate in detectable concentration until after hydrolysis of the DFP formed by reaction with $O_2$, as indicated by decay of the transient 650 nm absorbance prior to detection of the MVFC by EPR. The MVFC species accumulated rapidly between 0.25 and 1.0 s post-mixing, with a maximum observed concentration of ~70 μM (15% of protein monomers). We therefore conclude that the MVFC that is stabilized under limiting $O_2$ conditions results from a disproportionation of di-$Fe^{2+}$ and di-$Fe^{3+}$ FCs. It is not generated by direct $O_2$-driven oxidation of the di-$Fe^{2+}$ form because that intermediate is consumed by reaction with $O_2$ far more rapidly than remaining unreacted di-$Fe^{2+}$ centers.

Formation of the MVFC post-hydrolysis of the DFP species raises the possibility that the MVFC is produced by the reaction of remaining

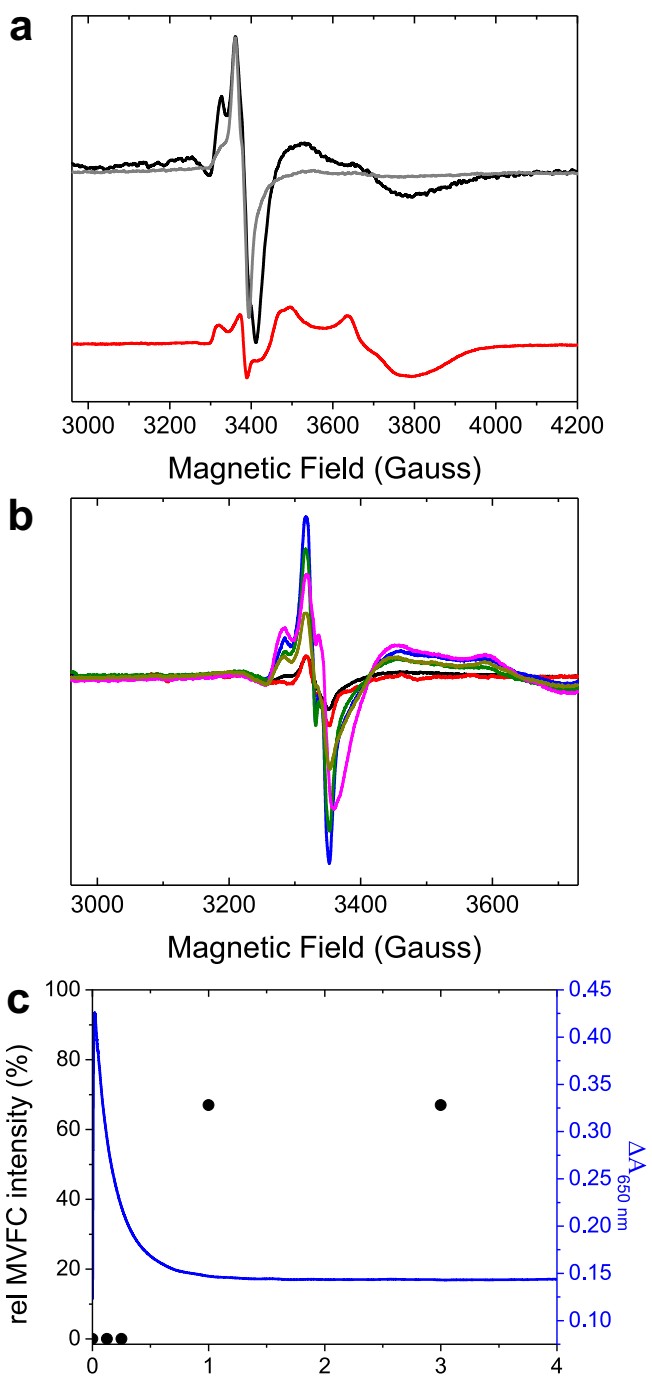

**Fig. 4 | O₂-limited reaction of FtMt. a** The EPR spectrum of 4.17 μM wild-type FtMt anaerobically incubated with 200 μM $Fe^{2+}$ then frozen 10 s after mixing with aerobic buffer, resulting in an O₂ concentration of 50 μM post-mixing (black trace). The red trace shows the MVFC formed by variant Y34F when O₂ is present in excess, and the gray trace shows the spectrum of wild-type protein when O₂ is present in excess for comparison. **b** Accumulation of the MVFC intermediate with increasing freezing time following the mixing of 20.8 μM wild-type FtMt with 1 mM $Fe^{2+}$ under aerobic conditions. Samples frozen 0.125 (black), 0.250 (red), 1 (blue), 3 (dark green), 6 (magenta) and 10 (dark yellow) s after mixing. **c** Comparison of the relative intensity of the MVFC of the samples from (b) (black circles, left axis) with the change in 650 nm absorbance as a function of time after mixing of an equivalent sample (blue trace, right axis). Source data are provided as a Source Data file.

di-$Fe^{2+}$ sites with the peroxide released during decay of the DFP form of the FC. However, control measurements (Figs. S8–S10) confirmed that the reaction of di-$Fe^{2+}$ FtMt FCs with peroxide is too slow to account for the observed rate of MVFC formation. Furthermore, reaction with peroxide only generates the MVFC on the wild-type protein when $Fe^{2+}$ is present in excess, whilst the O₂-driven reactions above would generate 250 μM peroxide together with 500 μM unreacted $Fe^{2+}$.

The impact of Ser at position 144 on the formation of MVFCs by FtMt under limiting O₂ was probed by using FtMt variant S144A and HuHF variant A144S. Whilst the S144A substitution resulted in the previously reported[28] increase in rate of iron oxidation and mineralization (Fig. S7), the corresponding A144S substitution in HuHF had a negligible effect on both processes (Fig. S7). Slow-freeze EPR revealed that, whilst the S144A substitution in FtMt led to a decrease in MVFC intensity of approximately 50%, no detectable MVFC was formed by HuHF variant A144S (Fig. S7).

## Products of O₂ reduction

Oxidation of di-$Fe^{2+}$ FCs to the mixed-valent form liberates a single electron that might be expected to cause the reduction of O₂ to superoxide. In the wild-type protein, Tyr34 has the potential to deliver a second reducing equivalent to O₂, or this could be derived from a second $Fe^{2+}$ ion at a remote FC, leading to the formation of peroxide in either scenario. Consistent with this, previous studies of the stoichiometry of the iron-O₂ chemistry catalyzed by the FC of FtMt indicated that two equivalents of $Fe^{2+}$ were oxidized per O₂ reduced[28].

In the case of variant Y34F, there is no oxidizable residue at position 34. However, a significant proportion of FCs in the mixed-valent state were observed following reaction with O₂, including when it was in excess. Thus, if FtMt FCs carry out their oxido-reductase chemistry in isolation from one another, this variant, which has no obvious source of a second reducing equivalent for O₂, would be expected to produce a higher proportion of superoxide as the product of O₂ reduction than the wild-type protein.

Production of peroxide during FtMt activity was assessed using the Amplex Red assay. Control experiments indicated that FtMt interferes with H₂O₂ detection, either by direct reaction or reaction with Complex I of horse radish peroxidase, preventing oxidation of the Amplex Red dye. Nevertheless, for wild-type and Y34F proteins, approximately 25% of the O₂ reduced was detected as H₂O₂ in the Amplex Red assay, suggesting that the Y34F substitution had no effect on the product of O₂ reduction.

In the case of variant Y34F, peroxide formation would necessitate either electron transfer between FCs, since each can provide only a single electron, or diffusion of superoxide between centers in order to acquire a second reducing equivalent. The latter would be expected to result in the observation of the superoxide adduct of the spin-trapping reagent DMPO. Ambient temperature EPR spectra of DMPO following incubation with wild-type or Y34F FtMt and 72 equivalents of $Fe^{2+}$ showed no evidence of superoxide formation (Fig. S11). Thus, we conclude that, in variant Y34F, and by extension in wild-type FtMt also, electron transfer from a remote FC provides a second reducing equivalent to FC-bound O₂, leading to peroxide as the product of O₂ reduction.

## Discussion

Both FtMt and the recently characterized cyanobacterial ferritin SynFtn contain a serine residue in a structurally equivalent position approximately 5 Å from the nearest iron (termed Fe2) of the FC. Iron oxidation catalyzed by SynFtn has been shown to proceed via an atypical $Fe^{2+}/Fe^{3+}$ MVFC intermediate in which only 50% of FC-bound iron is oxidized[30]. We therefore hypothesized that the presence of a similar intermediate in the catalytic cycle of FtMt may account for the previously reported unusual iron-binding and oxidation characteristics of this protein[28]. In fact, we were unable to reproduce the earlier reported

behavior of FtMt, which led to the conclusion that only half of the FCs were active[28]: in our hands, all of the FCs of FtMt were active, consistent with the recent structure-focused study of FtMt[41]. Despite this, our data do show that a MVFC intermediate accumulates on wild-type FtMt, but only under $O_2$-limiting conditions (of the type likely to occur in mitochondria), and that this is the result of disproportionation of di-$Fe^{2+}$ and di-$Fe^{3+}$ sites.

A recent computational study offered greater insight into the mechanism of SynFtn, suggesting that the role of the near-FC Ser residue in this protein was to stabilize superoxide bound to Fe2 following the one-electron reduction of $O_2$[31]. The mechanism also required population of a nearby cation binding site, whilst the generation of peroxide and 2 MVFCs as the product of $O_2$ reduction implied the presence of an inter-FC electron transfer route. The conserved tyrosine (Tyr34 in FtMt numbering) was essential to lower the activation energy for the reduction of $O_2$ bound to the di-$Fe^{2+}$ FC. FtMt also harbors a cation binding site that is not conserved in HuHF, around 10 Å from the FC[41]. Generation of peroxide as the product of $O_2$ reduction by FtMt variant Y34F and electron redistribution in the wild-type protein post-hydrolysis of the DFP species under limiting $O_2$ both imply the presence of an as yet unidentified inter-FC electron transfer route in FtMt that is absent in HuHF. We therefore propose that the catalytic cycle of FtMt, detailed in Fig. 5, is very similar to that of SynFtn but with a MVFC that is far more reactive toward $O_2$ than the di-$Fe^{2+}$ form. However, oxidation of the MVFC involves activation of $O_2$ by Tyr34 such that a significant proportion of variant Y34F becomes trapped in a metastable MVFC state. While direct oxidation to the DFP

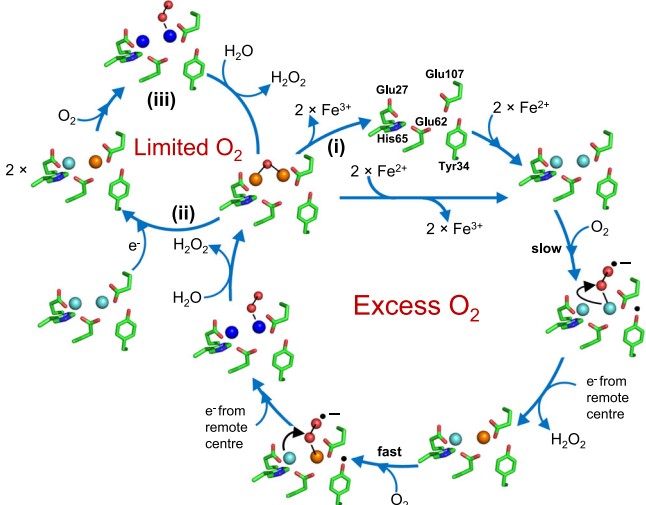

**Fig. 5 | Proposed mechanistic cycle of $Fe^{2+}$ oxidation by FtMt.** Following binding of two $Fe^{2+}$ ions to the apo FC (top of the main cycle), $O_2$ binds and is activated by transfer of a single electron from Tyr34 to yield a tyrosyl radical and a superoxo adduct of the di-$Fe^{2+}$ FC. The radical is quenched by electron transfer from a remote FC, and electron transfer from $Fe^{2+}$ at the site with superoxide bound yields the peroxide product of $O_2$ reduction and the MVFC intermediate. Binding of $O_2$ and its activation by the MVFC is rapid in comparison to that occurring at the di-$Fe^{2+}$ FC form, and hence, the MVFC intermediate does not accumulate under continuous turnover. This reaction results in the DFP intermediate, which is hydrolyzed to the metastable di-$Fe^{3+}$ form. In the presence of excess $O_2$, the metastable di-$Fe^{3+}$ FC breaks down to release $Fe^{3+}$ ions to the FtMt cavity, and the reaction cycle begins again (marked as (i)). Under limiting $O_2$, where both di-$Fe^{2+}$ and di-$Fe^{3+}$ FCs are present, a disproportionation reaction occurs via long-range electron transfer, resulting in two MVFCs (marked as (ii) on the smaller left-hand cycle). When $O_2$ is available again, these are oxidized to the di-$Fe^{3+}$ form (marked as (iii)). The overall stoichiometry of the reaction is identical to that catalyzed by cytosolic ferritins, with each center coupling the oxidation of two equivalents of $Fe^{2+}$ to $Fe^{3+}$ with the reduction of $O_2$ to peroxide. $Fe^{2+}$ ions are indicated in cyan, $Fe^{3+}$ are in brown, and the DFP is in royal blue.

state undoubtedly occurs in this variant, formation is slower than in the wild-type protein. We hypothesize that this is due to the energy required to form the DFP via a transient MVFC intermediate being raised by the substitution of Tyr34, to the extent that direct DFP formation becomes a competing reaction pathway. The major role of Ser144 is the stabilization of superoxide bound to Fe2 during the oxidative formation of the MVFC. The 50% reduction in observed MVFC upon substitution of Ser144 with Ala also showed that Ser at position 144 in FtMt favors disproportionation of di-$Fe^{2+}$ and di-$Fe^{3+}$ sites to generate MVFCs under $O_2$ limitation. Variant A144S of HuHF is unable to relax to the same MVFC state, presumably because the inter-FC electron transfer route required for FtMt function is missing or much less efficient in HuHF.

FtMt is implicated in oxidative stress response, but it seems unlikely that the role of the protein is reactive oxygen species (ROS) detoxification via a direct reaction. Peroxide reduction by FtMt is sluggish, results in damage to the protein, and it is known that the vast majority of peroxide detoxification in mitochondria is performed by other enzymes[42]. Similarly, superoxide detoxification is carried out by superoxide dismutase (SOD). Consumption of $O_2$ by the respiratory chain enzymes within the mitochondrial membrane means that $O_2$ concentration within the matrix is very low[43,44]. However, around 1% of the $O_2$ consumed during respiration is estimated to be converted to superoxide[45]. The action of SOD within the mitochondrial matrix will therefore result in a low but constantly replenished supply of $O_2$, the presumed substrate of FtMt. The physiological importance of the unusual iron oxidation mechanism of FtMt remains unclear, but one possibility is that the typical conditions within the mitochondrial matrix, low $O_2$ concentration combined with high iron flux, serve to poise the protein in the MVFC state. This form of the protein is highly reactive towards $O_2$ and may serve as a buffer against higher iron and fluctuating $O_2$ concentrations within the microaerobic environment of the mitochondrial matrix.

Strict conservation among mitochondrial ferritins of Ser at the position equivalent to Ser144 in the human protein (Fig. S1) supports the importance of this residue for their function within the mitochondrial matrix. Whilst cytosolic H-chain ferritins from some animals also contain Ser at the position equivalent to 144 in HuHF, these proteins exhibit considerable variation at this position, and our data suggest that the presence of Ser144 alone is not sufficient to influence FC mechanism[41]. However, even if it were to do so, the higher $O_2$ concentration and lower iron flux within the cytosol mean that accumulation of cytosolic H-chain in the MVFC state is unlikely to occur in vivo, and so the unusual mechanistic features of FtMt evolved specifically in the low $O_2$ environment of the mitochondrial matrix.

## Methods
### Protein expression and purification
Plasmids based on the pET21a expression vector and encoding FtMt, HuHF and Y34F (FtMt and HuHF), S144A (FtMt), and A144S (HuHF) variants were purchased from Genscript (UK). All FtMt constructs lacked the predicted mitochondrial targeting sequence and the first 9 amino acid residues at the N terminus of the mature protein[28]. Proteins were expressed from the *Escherichia coli* strain BL21 DE3. All cultures in liquid media were grown at 37 °C, 200 rpm shaking, and contained 100 µg mL$^{-1}$ ampicillin unless otherwise stated. Single colonies from LB agar plates containing 100 µg mL$^{-1}$ ampicillin, grown overnight at 37 °C, were picked into 5 mL liquid media (LB) and cultures were grown throughout the day. 400 µL of the resulting cell culture was used to inoculate 80 mL of LB, which was grown to saturation overnight. 50 mL of the saturated culture was diluted 1 part in 100 into 5 L of LB and grown until the optical density at 600 nm was in the range 0.6–0.8. Protein expression was induced by adding isopropyl β-d-1 thiogalactopyranoside (IPTG) to a final concentration of 100 µM. Cultures

were grown on for a further 20 h at 30 °C, 90 rpm shaking prior to harvesting by centrifugation.

Pellets were resuspended in 20 mM HEPES, 100 mM KCl, 0.1 mM EDTA, pH 7.8 (buffer A), and the cells disrupted by sonication. Debris was removed by centrifugation at $40,000 \times g$, 1 °C for 45 min. Thermally unstable proteins were precipitated from the supernatant by heating to 65 °C for 15 min and removed by a further round of centrifugation as above. Ferritin was precipitated from the supernatant by the addition of ammonium sulfate to a concentration of 0.55 g mL$^{-1}$. Precipitated protein was pelleted by a further round of centrifugation before redissolving in a minimum volume of buffer A and dialyzing against 1 L of identical buffer for a minimum of 12 h. Contaminating proteins were removed by size exclusion chromatography (HiPrep 26/60 Sephacryl S300HR, Cytiva) and contaminating DNA by anion exchange chromatography (HiTrap Q FF, Cytiva). For the latter, protein solutions were loaded in buffer A and eluted by stepping to 30% buffer B (20 mM HEPES, 100 mM KCl, 1 M NaCl, 0.1 mM EDTA, pH 7.8).

Protein as isolated contained small quantities of iron that was removed using the method of Bauminger et al[46]. Following iron removal, protein was exchanged into 100 mM MES pH 6.5 by centrifugation over a 10 kDa molecular weight cut-off cellulose membrane (Millipore). The absence of contaminating proteins was confirmed using SDS-PAGE, and ferritin was judged free from DNA contamination once the ratio of absorbance at 280 nm to 260 nm reached 1.5. Protein concentration was determined by absorbance, assuming $\varepsilon_{280\,nm} = 4.08 \times 10^5$ M$^{-1}$ cm$^{-1}$ for the 24meric protein cage[28].

In general, kinetic absorbance experiments were performed in triplicate for each of two independently prepared protein samples, which gave very similar results; representative data are shown. EPR slow-freeze experiments were repeated in duplicate, and a standard 10 s freeze spectrum following the addition of 72 Fe$^{2+}$ per protein was recorded at the beginning of each EPR experiment to ensure reproducibility between different protein preparations. For freeze-quench EPR, experiments were performed with a single protein preparation at increasing time points, resulting in a self-consistent set of spectra.

## Absorbance-monitored kinetic studies

Protein activity was monitored via the increase in absorbance at 340 nm resulting from the oxidation of Fe$^{2+}$ to Fe$^{3+}$, or by monitoring the transient absorbance at 650 nm resulting from the formation of a di-Fe$^{3+}$ peroxo (DFP) intermediate. The rate of mineral core formation was determined from the increase in 340 nm absorbance using a Hitachi U2900 spectrophotometer with the sample chamber maintained at 25 °C. Ferrous ammonium sulfate dissolved in 1 mM HCl was added to a final concentration of 200 μM to a 0.5 μM solution of FtMt in 100 mM MES, pH 6.5, in a 1 cm pathlength cuvette. The extinction coefficient of the mineral core was deduced from the net absorbance change once iron oxidation was complete. Iron oxidation at the FC was monitored by using an Applied Photophysics Bio-Sequential DX.17MV spectrophotometer with a 1 cm pathlength observation cell to mix equal volumes of 1 μM apo protein in 100 mM MES pH 6.5 and solutions of 6, 12, 18, 24, 30, 36, 42, 48, 60, 72 or 96 μM ferrous ammonium sulfate in 1 mM HCl. Data were collected using ProData SX software (Applied Photophysics). The time dependence of absorbance increases at 340 nm was fitted to the sum of two exponential processes, encompassing rapid (r) and slow (s) components, using OriginPro 8 (OriginLab):

$$\Delta A_{340}(t) = \Delta A_{340}^{(tot)} - \Delta A_{340}^{r} e^{-k_r t} - \Delta A_{340}^{s} e^{-k_s t} \qquad (1)$$

Formation of the DFP intermediate was also detected using the Applied Photophysics Bio-Sequential DX.17MV spectrophotometer with ProData SX, but required an increase in protein concentration to 8 μM (192 μM in monomer) prior to mixing with 384 μM Fe$^{2+}$ (2 Fe$^{2+}$ per FC). The rate of increase and subsequent decrease in 650 nm

absorbance was used to deduce the rate of formation and decay of the DFP intermediate, whilst the increase in absorbance at 340 nm was used to deduce the overall rate of Fe$^{2+}$ oxidation.

## Electron paramagnetic resonance

EPR spectra were recorded using Bruker EMX and Bruker E500 EPR spectrometers (both X-band) with WinEPR/XEPR software (Bruker BioSpin). The low-temperature EPR spectra were measured at 10 K using Oxford Instruments liquid helium systems. Instrument parameters for low-temperature EPR measurements were as follows: microwave frequency $\nu_{MW} = 9.467$ GHz, modulation frequency $\nu_M = 100$ kHz, time constant $\tau = 82$ ms, microwave power = 3.19 mW or 0.05 mW, modulation amplitude $A_M = 5$ G or 3 G (for narrower field range), scan rate $\nu = 22.6$ Gs$^{-1}$ or 6.0 Gs$^{-1}$ (for narrower field range). Wilmad SQ EPR tubes (Wilmad Glass, Buena, NJ) with OD = 4.05 ± 0.07 mm and ID = 3.12 ± 0.04 mm (mean ± range) were used for low-temperature measurements. Two methods of sample freezing were used.

Protein samples in EPR tubes were mixed with the appropriate volume of a 25 mM stock Fe$^{2+}$ solution and frozen 10 s thereafter by plunging the tubes into methanol cooled with solid CO$_2$ (slow freezing method). To produce samples sub-stoichiometric in O$_2$, protein solutions were rendered anaerobic by stirring under an argon atmosphere for 30 min prior to incubation with Fe$^{2+}$. Reactivity was then initiated by mixing with buffer containing 250 μM dissolved O$_2$ prior to freezing as described above. The addition of an appropriate volume of a 50 mM stock H$_2$O$_2$ solution in place of oxygenated buffer to equivalent samples was used to generate samples with H$_2$O$_2$-driven oxidation of Fe$^{2+}$.

Samples frozen at times <10 s (the Rapid Freeze-Quench technique, RFQ) were prepared by mixing equal volumes of an 8.33 μM protein solution (200 μM in subunit monomer) and a 0.6 mM Fe$^{2+}$ solution in a set-up combining an Update Instrument Syringe Ram Controller (Madison, WI) and a home-built apparatus for freezing the ejected mixtures on the surface of a rotating aluminum disk kept at liquid nitrogen temperature, as described elsewhere[47]. Final protein concentration was 4.17 μM (100 μM in monomer). Proteins were in 100 mM MES, pH 6.5, and Fe$^{2+}$ solutions in 50 mM HCl (slow freezing) or 1 mM HCl (RFQ). The concentration of the MVFC species in various samples was determined by comparison of the integrated intensity of the associated EPR signal with that of a 100 μM CuEDTA standard.

The ambient temperature EPR experiments on radical trapping with the spin trap reagent 5,5-dimethyl-1-pyrroline-N-oxide (DMPO) were performed with the use of a Bruker AquaX system for liquid sample measurements. To minimize the dead volume of AquaX, the inlet tubing was shortened to 1 cm. 200 μL samples were prepared in 100 mM MES pH 6.5 with the following final concentrations: 80 mM DMPO, 12 μM FtMt, and 288 μM Fe$^{2+}$ and immediately loaded by syringe into the AquaX capillaries system. Spectra were acquired using the following parameters: microwave frequency $\nu_{MW} = 9.84$ GHz, modulation frequency $\nu_M = 100$ kHz, time constant $\tau = 51$ ms, microwave power = 0.63 mW, modulation amplitude $A_M = 1$ G, scan rate $\nu = 1.9$ Gs$^{-1}$.

## Fluorescence emission spectroscopy

Spectra were recorded on an Edinburgh Instruments FS5 spectrofluorometer equipped with a 96-well plate reader. Hydrogen peroxide was detected using an Amplex Red kit (Thermo Fisher) with excitation at $\lambda = 530$ nm and emission scans across the range 550–650 nm. Fluorescence intensity at 590 nm following the aerobic oxidation of 9.6 μM Fe$^{2+}$ by 0.2, 0.4, 0.6, 0.8 or 1.0 μM wild-type or Y34F FtMt was used to quantify released H$_2$O$_2$ by comparison to a plot generated from standard peroxide solutions of 1–5 μM.

## Reporting summary

Further information on research design is available in the Nature Portfolio Reporting Summary linked to this article.

## Data availability

All data supporting the findings of this study are available within the paper and its Supplementary Information. Source data are provided with this paper. Expression constructs are available upon request from Nick Le Brun (n.le-brun@uea.ac.uk). Source data are provided with this paper.

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

## Acknowledgements

The authors wish to thank Dr Andrew Gates (UEA) for access to the stopped-flow instrument. This work was supported by the UK's Biotechnology and Biological Sciences Research Council through grants BB/R002363/1 (J.M.B. and N.L.B.), BB/R003203/1 (D.A.S.) and BB/T01802X/1 (D.A.S.) and by UEA.

## Author contributions

J.M.B. and N.L.B. conceived the study. J.M.B., Z.B., J.P. and D.A.S. performed the experiments. J.M.B., Z.B. and D.A.S. analyzed the data. J.M.B., D.A.S., G.R.M., and N.L.B. wrote the manuscript.

## Competing interests

The authors declare no competing interests.
