## [Transparent Peer Review file · Nature Communications]

Human mitochondrial ferritin exhibits highly unusual iron-O₂ chemistry distinct from that of cytosolic ferritins

Corresponding Author: Professor Nick Le Brun

Version 0:

Reviewer comments:

Reviewer #1

(Remarks to the Author)

Bradley et al. reported that Fe²⁺ oxidation at the ferroxidase center of FtMt involves radical formation on a conserved Tyr residue (Tyr34). FtMt also forms a mixed-valent ferroxidase center (MVFC) under the O₂-limiting conditions. Although this manuscript provides some new insights into the different behaviors of FtMt and HuHF, the functional significance and underlying mechanisms remain largely undefined.

1. The major differences in the iron binding and oxidation properties between FtMt and HuHF have already been reported by previous studies (References 28 and 22). Additionally, the conserved Tyr residue of the FC in all H-chain type ferritins (Tyr34 in FtMt and HuHF) has also been shown to play an important role in Fe²⁺ oxidation and mineralization (References 17, 36-38). It would be important to make a direct comparison between FtMt and HuHF on Fe²⁺ oxidation and mineralization using both wild-type and Y34F variant.

2. The authors detected a MVFC in FtMt, but not in HuHF, under the O₂-limiting conditions. The Y34F variant of FtMt also forms a MVFC with excess O₂. However, the physiological relevance of the MVFC intermediate in FtMt is unclear.

3. "The highly unusual iron-O₂ chemistry exhibited by FtMt shows how subtle structural differences around the catalytic centers of the mitochondrial and cytosolic proteins have fundamental mechanistic consequences" was not supported by experimental data. Although the authors proposed that Ser144 might underlie the observed differences between FtMt and HuHF, whether Ser144 in FtMt is important for the MVFC formation needs to be further determined. Moreover, does mutation of the equivalent position in HuFH to Serine render it to form MVFCs under O₂ limitation?

Reviewer #2

(Remarks to the Author)

This paper looks in detail at the mechanism of Fe²⁺ oxidation of a mitochondrial ferritin (MtFt). MtFt is differentially expressed in different tissues. Its expression is not under control of the normal Iron Responsive Element, and mis-regulation of MtFt occurs in several diseases and neurological disease states. This makes MtFt a very attractive target for study.

The paper looks in detail at the oxidative reaction of MtFT with Fe²⁺ and O₂. The kinetic and EPR are beautifully presented and provide evidence for the involvement of a Tyr residue. A mechanism for FtMt in which the ferroxidase centre drives Fe²⁺ oxidation at low and high O₂ conditions is presented in Figure 6, and is distinct from previous mechanisms for the closely related cytosolic ferritins. The key points are that a mixed valent (Fe²⁺/Fe³⁺) ferroxidase centre is formed under low O₂ (which is relevant to the low-oxygen conditions in mitochondria), and a conserved Tyr radical is involved.

The discovery that cytosolic and FtMt have fundamentally different mechanisms is, to my reading, an important contribution because of the potential relevance to disease. I am very supportive of the paper for this reason. But the paper is very long, and so I would ask the authors to condense it in places (Results), and to focus the Discussion a little more, so that the main messages stand out for a wide read

Version 1:

Reviewer comments:

Reviewer #1

(Remarks to the Author)

1. My concerns regarding the novelty of this study has not been adequately addressed by the revision. Major differences in iron binding and oxidation properties between FtMt and HuHF have already been reported previously (see ref 22, 28). Although the authors proposed MVFC as an alternative mechanism to explain these differences, the causal relationship needs to be further demonstrated with new experimental data.

2. The physiological relevance of MVFC in FtMt was not addressed by the revised manuscript. Given HuHF can function without MVFC, it would be important to further validate that MVFC in FtMt is indeed formed under physiological conditions in the mitochondria and its disruption will have detectable functional consequences.

3. The authors proposed that Ser144 might underlie the observed differences between FtMt and HuHF, but this was not supported by the new data in Figure S7 – MVFC was only slightly diminished but not abolished in S144A of FtMt, and A144S of HuHF failed to form MVFC. Therefore, the mechanistic underpinnings of MVFC formation remain unclear.

Version 2:

Reviewer comments:

Reviewer #2

(Remarks to the Author)

I am happy with the authors response to the requests of the reviewers. I do think the paper offers an advance over previous work and I agree with their professional and diplomatic approach of not being overly critical of previous papers. It is not clear what reviewer 1 means when they asked experiments to demonstrate a causal relationship are not clear. So in the absence of specific information of precisely what experiments are needed, then the authors cannot address that point. The suggestion of experiments in mitochondria is totally unrealistic - that could be a years-long project on its own - well beyond the scope of a mechanistic study like the current paper, and not a reasonable request from the reviewer. The author's interpretation of a 50% drop in activity (which is significant) for the S144 mutant is sensible.

From my point of view the authors have sensibly adjusted the paper and I am happy to see it published.

Reviewer #3

(Remarks to the Author)

I reviewed this manuscript, especially focusing on a disagreement between the authors and Reviewer #1. Although most studies about FtMt have been focused on the physiological consequences of misregulation of FtMt expression, the biochemical and biophysical properties of the protein and mechanisms of Fe²⁺ oxidation/mineralization remain relatively poorly characterized. In this study, authors have shown the novelty in terms that FtMt has different iron binding and oxidation properties from cytosolic ferritin. Although Reviewer #1 concerned about the differences in iron binding and oxidation properties between the present study and previous studies (ref 22, 28), authors were unable to reproduce the earlier reported behavior of FtMt (ref 28), as described in Discussion. In addition, authors showed that all of the FCs of FtMt were active, consistent with the recent structure-focused study of FtMt, and that a MVFC intermediate accumulates on wild-type FtMt only under O₂-limiting conditions, resulted from disproportionation of di- Fe²⁺ and di- Fe³⁺ sites. In these respects, authors have provided newly uncovered iron binding and oxidation properties of FtMt, and adequately addressed the reviewer's concerns.

Overall the contents of manuscript are specialized and may be somewhat difficult to understand methods and interpretation of results for general readers, but the results seem to have an impact on this field.

February 4, 2025

Response to reviewers' comments

Reviewer #1:

Bradley et al. reported that Fe²⁺ oxidation at the ferroxidase center of FtMt involves radical formation on a conserved Tyr residue (Tyr34). FtMt also forms a mixed-valent ferroxidase center (MVFC) under the O₂-limiting conditions. Although this manuscript provides some new insights into the different behaviors of FtMt and HuHF, the functional significance and underlying mechanisms remain largely undefined.

1. The major differences in the iron binding and oxidation properties between FtMt and HuHF have already been reported by previous studies (References 28 and 22). Additionally, the conserved Tyr residue of the FC in all H-chain type ferritins (Tyr34 in FtMt and HuHF) has also been shown to play an important role in Fe²⁺ oxidation and mineralization (References 17, 36-38). It would be important to make a direct comparison between FtMt and HuHF on Fe²⁺ oxidation and mineralization using both wild-type and Y34F variant.

- Whilst differences in the iron binding and oxidation properties of FtMt and HuHF have indeed been reported previously, these data were not interpreted as reflecting differences in the mechanism of iron oxidation, but rather that half of FtMt ferroxidase centres are inactive. It was suggested that negative cooperativity of iron binding at remote ferroxidase centres, mediated by Ser144, was responsible for the unusual properties of FtMt (ref 28). However, our data show that all ferroxidase centres in FtMt are active, and thus negative cooperativity is not the root of the different behaviours of the two proteins. Instead, these arise from mechanistic differences that result in the observation of a MVFC under O₂-limiting conditions, representing a catalytic intermediate in this protein that is not formed by HuHF. This hypothesis is supported by the formation of a metastable MVFC by FtMt variant Y34F when O₂ is present in excess. No such species is formed by HuHF variant Y34F, indicating that the mechanism of iron oxidation at the ferroxidase centre of the two proteins differs significantly despite their considerable sequence identity in this region. This mechanistic variation was not identified in the previous studies (refs 22, 28, 41), and the data reported in this new manuscript explain the origin of the reported differences between FtMt and HuHF.

Since the different consequences of the Y34F substitutions in FtMt and HuHF are key to the proposed mechanism of iron oxidation by FtMt, the manuscript has been re-written to include direct comparison of the kinetics of this process between wild-type FtMt, wild-type HuHF and the Y34F variants of each. We thank the reviewer for this suggestion to strengthen the manuscript. In line with the request from Reviewer 2 that the manuscript be shortened where possible, the additional data are presented in Supplementary Information.

2. *The authors detected a MVFC in FtMt, but not in HuHF, under the O₂-limiting conditions. The Y34F variant of FtMt also forms a MVFC with excess O₂. However, the physiological relevance of the MVFC intermediate in FtMt is unclear.*

- We thank the reviewer for this comment. We have demonstrated that the MVFC is formed and stable in FtMt under O₂-limiting conditions, such as those that have been shown to exist in the mitochondrial matrix (refs 43, 44). We also now present new data (in Figure S6) showing that this form of the protein is highly reactive towards newly introduced O₂, suggesting a possible physiological function in buffering against fluctuating O₂ levels. While demonstrating physiological function is difficult, we think it is fair to conclude that the MVFC is physiologically relevant, i.e. is likely to be formed under physiological conditions of iron/O₂ in the mitochondrion.

3. *“The highly unusual iron-O₂ chemistry exhibited by FtMt shows how subtle structural differences around the catalytic centers of the mitochondrial and cytosolic proteins have fundamental mechanistic consequences” was not supported by experimental data. Although the authors proposed that Ser144 might underlie the observed differences between FtMt and HuHF, whether Ser144 in FtMT is important for the MVFC formation needs to be further determined. Moreover, does mutation of the equivalent position in HuFH to Serine render it to form MVFCs under O₂ limitation?*

- We thank the reviewer for this comment and agree that the original data did not demonstrate that Ser144 was the cause of the observed difference between FtMt and HuHF. The section of the abstract in question has been edited to reflect this. As before, we note that the significant degree of sequence identity between FtMt and HuHF in the vicinity of the catalytic centre does not result in identical mechanism of iron oxidation in the two proteins, and this is the point we emphasise in the revised manuscript accompanied by additional data. Further data has been added to Figure S7 of the manuscript detailing the impact of substitutions at position 144 of both HuHF and FtMt on iron oxidation kinetics and MVFC formation under limiting O₂. MVFC formation was diminished (though not abolished) in variant S144A of FtMt, consistent with an important role for Ser144 in stabilising the MVFC form under limiting O₂. Variant A144S of HuHF did not form the MVFC under limiting O₂. Thus, introducing a Ser residue here in HuHF is alone not sufficient for MVFC formation, most likely because disproportionation of di-Fe²⁺ and di-Fe³⁺ sites is also required under limiting O₂ (Figure 5), involving long-range electron transfer between FCs in different subunits. This occurs apparently occurs readily in FtMt, but not in HuHF.

Reviewer #2:

This paper looks in detail at the mechanism of Fe²⁺ oxidation of a mitochondrial ferritin (MtFt). MtFt is differentially expressed in different tissues. Its expression is not under control of the normal Iron Responsive Element, and mis-regulation of MtFt occurs in several diseases and neurological disease states. This makes MtFt a very attractive target for study.

The paper looks in detail at the oxidative reaction of MtFT with Fe²⁺ and O₂. The kinetic and EPR are beautifully presented and provide evidence for the involvement of a Tyr residue. A mechanism for FtMt in which the ferrocyclase centre drives Fe²⁺ oxidation at low and high O₂ conditions is presented in Figure 6, and is distinct from previous mechanisms for the closely related cytosolic ferritins. The key points are that a mixed valent (Fe²⁺/Fe³⁺) ferrocyclase centre is formed under low O₂ (which is relevant to the low-oxygen conditions in mitochondria), and a conserved Tyr radical is involved.

The discovery that cytosolic and FtMt have fundamentally different mechanisms is, to my reading, an important contribution because of the potential relevance to disease. I am very supportive of the paper for this reason. But the paper is very long, and so I would ask the authors to condense it in places (Results), and to focus the Discussion a little more, so that the main messages stand out for a wide read.

- We thank the reviewer for their positive comments on the manuscript and were pleased they approved of the presentation of the kinetic and spectroscopic data. In line with their request, we have significantly shortened the manuscript, while also adding new data in response to the points raised by Reviewer 1. Data pertaining to the peroxide-driven oxidation of iron bound to FtMt has been moved to Supplementary Information, where it is more appropriately located because the primary motivation for inclusion was to demonstrate that peroxide released during the hydrolysis of the DFP form of the FC was not the source of MVFC formation under limiting O₂. Furthermore, the Discussion section of the manuscript has been re-written to provide a more focused consideration of only the main conclusions from the data.

March 5, 2025

Response to reviewer#1's comments

1. My concerns regarding the novelty of this study has not been adequately addressed by the revision. Major differences in iron binding and oxidation properties between FtMt and HuHF have already been reported previously (see ref 22, 28). Although the authors proposed MVFC as an alternative mechanism to explain these differences, the causal relationship needs to be further demonstrated with new experimental data.

We are not in the habit of unnecessarily criticising the work of other researchers: it's not civil and you never know who is reviewing your manuscript. The reviewer points to the earlier work described in refs 22 and 28 as having already demonstrated major differences between FtMt and HuHF. As we point out in the manuscript (while also trying not to be overly critical), a significant part of the data reported in those two papers is not reproduced in our manuscript. On page 6, we wrote:

In contrast to early studies of FtMt conducted at pH 7.0²⁸, the above data indicated the rapid oxidation of 48 equivalents of Fe²⁺ per protein cage.

Thus, the most important observations reported from those initial studies are actually wrong: Half of the 24 ferroxidase centres of FtMt are not inactive and the protein is not significantly less active than HuHF. This was also pointed out in the recent structural study by Ciambellotti and co-workers (ref 41). They also failed to reproduce the behaviour reported in the earlier studies:

'At variance with previous reports on hMTF,[22] our kinetic data in solution are consistent with a full set of 24 active ferroxidase centers in hMTF, that give rise to the transient formation of diferric-peroxo intermediates that then evolve into ferric oxo species on a similar time scale as observed for homopolymeric recombinant HuHf'.

In the newly-revised version of the manuscript, we try to make the point clearer that this key aspect of the earlier data was not reproduced by us or another independent group.

Despite this, it is the case that FtMt does behave differently to HuHF. That is what we set out to understand. We have carried out a detailed in vitro study of the mechanism of Fe²⁺ oxidation catalysed by FtMt, demonstrating the production of a MVFC that is stable at low ratios of O₂ to Fe²⁺, and is formed by electron redistribution between (remote) ferroxidase centres. This is

entirely novel for any ferritin (i.e, distinct from the only other reported MVFC, in a bacterial ferritin), revealing the basis of the mechanistic differences between FtMt and HuHF.

The reviewer states '*Although the authors proposed MVFC as an alternative mechanism to explain these differences, the causal relationship needs to be further demonstrated with new experimental data.*'

We have no idea what this means. We have not proposed the formation of a MVFC as an alternative mechanism; the earlier proposed mechanism in which half of the FCs are inactive was in error, and as a consequence, the earlier work does not really offer any mechanistic insight beyond noting that Ser144 is likely somehow important. The study we have undertaken, as outlined above, provides the first comprehensive understanding of the FtMt catalytic mechanism, which involves the novel formation of a MVFC species.

2. The physiological relevance of MVFC in FtMt was not addressed by the revised manuscript. Given HuHF can function without MVFC, it would be important to further validate that MVFC in FtMt is indeed formed under physiological conditions in the mitochondria and its disruption will have detectable functional consequences.

We have demonstrated that a MVFC is generated in FtMt under low O₂ conditions, which are known to exist in the mitochondrion matrix. Thus, it follows that it is highly likely to be physiologically relevant. We acknowledge that it remains unclear why FtMt utilises this different mechanism. While we have discussed this, definitively addressing that question will be the subject of future studies. The reviewer will be fully aware that it is not practically possible to do what they're asking: simply detecting FtMt expression in vivo is challenging so detection of relatively weak MVFC EPR signals in whole cells (or isolated mitochondria) would not be possible because of background signals.

3. The authors proposed that Ser144 might underlie the observed differences between FtMt and HuHF, but this was not supported by the new data in Figure S7 – MVFC was only slightly diminished but not abolished in S144A of FtMt, and A144S of HuHF failed to form MVFC. Therefore, the mechanistic underpinnings of MVFC formation remain unclear.

This point concerns Ser144 and the proposal that, as one of the few obvious differences in the vicinity of the ferroxidase centres of the two proteins, it might be important for the mechanistic differences between FtMt and HuHF. The reviewer concludes that the proposal is not supported by the data. In fact, we report a 50% reduction in MVFC on substitution of Ser144 by Ala. It is naïve to imagine that the entire basis of the mechanistic difference is this single residue difference; indeed, we pointed out in the manuscript that some cytosolic H-chain ferritins in other mammals feature a Ser at the 144 position, so it is clearly more complex and there must be other subtle differences that also contribute, a point also made by Ciambellotti *et al* in ref 41. In the newly revised version of the manuscript, we try to make this point clearer.